# Luciferase-LOV BRET enables versatile and specific transcriptional readout of cellular protein-protein interactions

Christina K Kim[1], Kelvin F Cho[2], Min Woo Kim[1], Alice Y Ting[1,3,4]*

[1]Department of Genetics, Stanford University, Stanford, United States; [2]Cancer Biology Program, Stanford University, Stanford, United States; [3]Department of Biology, Stanford University, Stanford, United States; [4]Chan Zuckerberg Biohub, San Francisco, United States

**Abstract** Technologies that convert transient protein-protein interactions (PPIs) into stable expression of a reporter gene are useful for genetic selections, high-throughput screening, and multiplexing with omics technologies. We previously reported SPARK (Kim et al., 2017), a transcription factor that is activated by the coincidence of blue light and a PPI. Here, we report an improved, second-generation SPARK2 that incorporates a luciferase moiety to control the light-sensitive LOV domain. SPARK2 can be temporally gated by either external light or addition of a small-molecule luciferin, which causes luciferase to open LOV via proximity-dependent BRET. Furthermore, the nested 'AND' gate design of SPARK2—in which both protease recruitment to the membrane-anchored transcription factor and LOV domain opening are regulated by the PPI of interest—yields a lower-background system and improved PPI specificity. We apply SPARK2 to high-throughput screening for GPCR agonists and for the detection of trans-cellular contacts, all with versatile transcriptional readout.
DOI: https://doi.org/10.7554/eLife.43826.001

*For correspondence:
ayting@stanford.edu

Competing interests: The authors declare that no competing interests exist.

## Introduction

Because protein-protein interactions (PPIs) are central to every biological signaling pathway, there has been keen interest in developing ever-more specific, sensitive, and versatile reporters for detecting them in living cells. In general, existing tools to detect PPIs fall into two broad categories: real-time reporters such as those based on FRET (*Truong and Ikura, 2001*) or protein complementation assays (*Kerppola, 2006*), and transcriptional reporters such as yeast two-hybrid (*Miller and Stagljar, 2004*) and TANGO (*Barnea et al., 2008*). The two categories are complementary in that the former reads out PPI dynamics, while the latter provides a stable 'memory' of a previous PPI that can be used for cell-based selections or high-throughput screening. We previously reported 'SPARK', for 'Specific Protein Association tool giving transcriptional Readout with rapid Kinetics', a transcriptional PPI reporter that improves upon yeast two-hybrid and TANGO designs by incorporating light-gating. As shown in *Figure 1A*, transcriptional activation requires *both* a PPI event and the presence of blue light. Not only does the light-gating dramatically reduce the background of the system, but it also enables temporal specificity because the user can sync the light window to the biological time period of interest (e.g. just before or just after addition of a drug). Instead of integrating PPI events over many hours or days, as yeast two-hybrid and TANGO do, SPARK integrates PPI events over just a 5 min user-selected time window determined by exogenous blue light delivery.

In working with first-generation SPARK (SPARK1), however, we noticed two limitations. The first is that, due to its intermolecular design, SPARK1 performance is highly expression-level dependent. As shown in *Figure 1A–B*, protein A-protein B interaction drives together the low-affinity tobacco etch

virus protease (TEVp) and its recognition sequence (TEVcs), leading to cleavage and transcription factor (TF) release when light is also present. When SPARK1 components are expressed at high levels, however, TEVp interaction with TEVcs becomes proximity-*in*dependent, that is there is significant cleavage and TF release even when A-B interaction does *not* occur (likely because SPARK1 component levels exceed or are close to the TEVp-TEVcs $K_m$ which is estimated to be ~450 μM (*Kapust et al., 2001*; *Kapust et al., 2002*); *Figure 1C*). Our practical observation is that SPARK1 performance is best when the components are expressed stably or via viral transduction. Transient transfection methods such as lipofectamine or PEI result in higher expression levels and consequently high background. This reduction in specificity limits the robustness and therefore utility of the tool.

The second limitation is that light delivery presents a technical challenge, particularly for high-throughput screens, as many robotic platforms are not equipped to dark-cage, deliver light, and then dark-cage again multiwell plates at specified timepoints. Instead, these devices are optimized for liquid handling, that is delivery of compounds to multiwell plates is much more straightforward than controlled delivery of light.

To address both these challenges, we have developed second-generation SPARK (SPARK2), which incorporates a luciferase to control the light, oxygen, and voltage sensitive (LOV) domain, most likely via a BRET-type mechanism (Bioluminescence Resonance Energy Transfer; *Figure 1D*). Luciferases are enzymes that use $O_2$ and a small molecule 'luciferin' substrate to catalyze the generation of light. First, by utilizing a luciferase, we find that it is possible to time-gate SPARK by *either* luciferin addition or exogenous blue light delivery. This improves the versatility of the tool and makes it more compatible with high-throughput screening platforms. Second, because we found that luciferase-LOV BRET is strongly proximity-dependent, we made LOV domain uncaging also dependent on protein A-protein B interaction by fusing the luciferase to the protease of SPARK2, rather than directly to the LOV domain (*Figure 1E*). Whereas in SPARK1, only the TEVp recruitment to TEVcs was regulated by A-B interaction, now in SPARK2 the LOV domain is *also* regulated by A-B interaction. We find that this 'double dependence' on A-B interaction makes SPARK2 a much more specific reporter of cellular PPIs, even at high expression levels (*Figure 1F*). We demonstrate the utility of SPARK2 on multiple G-protein-coupled receptors (GPCRs), detecting their agonist-dependent activation and subsequent arrestin recruitment with signal to noise ratios as high as 20-fold. We then extend the SPARK2 design to detection of *trans*-cellular interactions.

## Results

### Selection of NanoLuc as the luciferase donor for LOV

First-generation SPARK1 incorporates an evolved photosensory domain from *Avena sativa* (asLOV2) to regulate access to the protease cleavage site (TEVcs, *Figure 1B*). The LOV domain contains a flavin chromophore that covalently reacts with cysteine 450 upon irradiation with 450 nm blue light. This in turn drives a large conformational change of LOV's C-terminal J-alpha helix (*Harper et al., 2003*), which can expose or conceal recognition sites, such as TEVcs.

A recent report describes that cell death can be mediated by the action of luciferase on the flavin-containing protein miniSOG (*Proshkina et al., 2018*), suggesting that luciferase-catalyzed uncaging of asLOV2 may also be possible. To explore this possibility, we considered the range of luciferases used for biological research; these include *Vargula hilgendorfii* (VLuc [*Maguire et al., 2013*; *Thompson et al., 1989*]), *Renilla reniformis* (RLuc [*Saito et al., 2012*; *Takai et al., 2015*; *Matthews et al., 1977*]), *Gaussia princeps* (GLuc [*Verhaegent and Christopoulos, 2002*; *Welsh et al., 2009*; *Kim et al., 2011a*; *Berglund et al., 2013*]), and *Oplophorus gracilirostris* (Nano-Luc [*Hall et al., 2012*; *Stacer et al., 2013*]). Whereas VLuc requires ATP to produce bioluminescence, limiting it to intracellular use, the others do not (*Tung et al., 2016*). Among the ATP-independent variants, NanoLuc has been characterized as one of the brightest-emitting luciferase variants (*England et al., 2016*), and its emission wavelength (450 nm) overlaps with the activation spectrum of LOV. NanoLuc's small size (19 kDa) also reduces potential steric problems when fused to other proteins, and its substrate, furimazine, exhibits greater stability than the coelenterazine

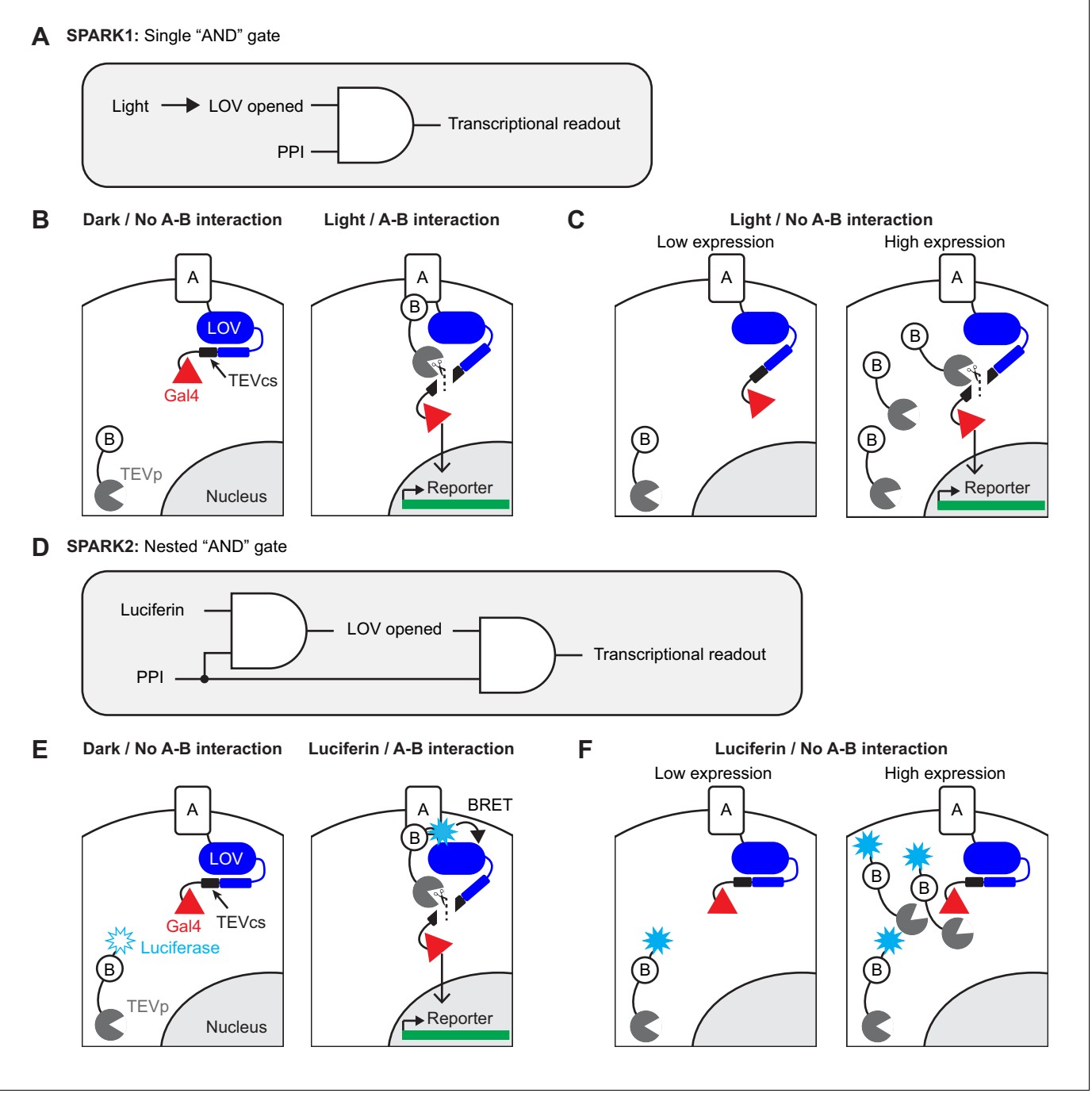

**Figure 1.** Motivation and design for SPARK2. (**A**) Logic diagram for single 'AND' gate used in first-generation SPARK1 (*Kim et al., 2017*). Transcription of the reporter gene requires both light (to open the LOV domain) AND a protein-protein interaction (PPI). (**B**) Schematic of SPARK1. Protein A is fused to the LOV domain, protease cleavage site (TEVcs), and transcription factor (TF, Gal4). Protein B is fused to a low-affinity variant of the protease TEVp. In the absence of an interaction between proteins A and B, TEVp recognition of and binding to TEVcs is minimal. Furthermore in the absence of light, the LOV domain cages the TEVcs, protecting it from spurious cleavage by the TEVp. If both light and an A-B interaction are present, then TEVp is recruited to the exposed TEVcs, resulting in the release of the TF Gal4 to the nucleus where it can drive expression of an exogenous reporter gene. (**C**) Schematic of PPI-independent background at high SPARK1 expression levels. If protein B-TEVp levels are sufficiently high, TEVp cleavage of the TEVcs could occur even in the absence of protein A-protein B interaction, when light is present. (**D**) Logic diagram for nested 'AND' gate used in second-generation SPARK2. Both TEVp recruitment to TEVcs *and* LOV domain opening (via Luciferase-LOV BRET) require a PPI. This double filter for PPI increases the specificity of SPARK2 for the PPI of interest and reduces PPI-independent background. (**E**) Schematic of SPARK2. A blue light-emitting luciferase is fused to protein B and TEVp. Instead of externally supplied light, the luciferase's substrate luciferin is supplied as a small-molecule drug. When there is an A-B interaction and luciferin is present, the luciferase activates the LOV domain via BRET, allowing the TEVp to cleave the now-

*Figure 1 continued on next page*

*Figure 1 continued*

accessible TEVcs. (**F**) Schematic of the reduction in background signal in the Luciferin/No PPI condition with SPARK2. Even if the luciferase-arrestin-TEVp is expressed at higher levels, TEVp cleavage of the TEVcs does not occur in the Luciferin/no PPI condition, as the LOV domain remains caged.

DOI: https://doi.org/10.7554/eLife.43826.002

analogues used for RLuc, GLuc, and sbLuc (*Hall et al., 2012*). For these reasons, we selected Nano-Luc as our luciferase donor to LOV.

## NanoLuc-LOV BRET is efficient and proximity-dependent

To determine whether NanoLuc is able to activate the LOV domain, we first used the LOVTRAP assay (*Wang et al., 2016*) in combination with a simple imaging-based readout. In LOVTRAP, an engineered peptide Zdk1 binds to the C-terminus of the LOV domain in the dark. Upon blue light irradiation and LOV conformational change, the Zdk1 peptide diffuses away from the LOV domain. The system is reversible, such that termination of blue light results in Zdk1 rebinding to LOV. We fused NanoLuc between β2AR and LOV (β2AR-NanoLuc-LOV) and expressed the construct in HEK293T cells together with an mCherry-Zdk1 fusion protein (*Figure 2A*). At baseline in the dark, we observed mCherry localization at the cell membrane, but in response to brief exposure to Nano-Luc's substrate, furimazine (10 µM for 1 min), or to blue light (10 s), the mCherry rapidly relocalized into the cytosol (*Figure 2B–C*). Cytosolic mCherry fluorescence was quantified as the average fluorescence 0.5 to 1.5 µm adjacent to the membrane (dashed boxes in *Figure 2C*). We observed comparable amounts of mCherry relocalization to cytosol between furimazine and blue light conditions (*Figure 2D*), suggesting that the NanoLuc-LOV activation was as efficient as exogenous blue light-LOV activation for releasing Zdk1.

To test whether NanoLuc activation of LOV is proximity-dependent, we prepared a construct in which NanoLuc was fused to the extracellular N-terminus of β2AR instead (*Figure 2E*; NanoLuc-β2AR-LOV), and we compared this side-by-side to the intracellular NanoLuc fusion construct from *Figure 2A* (β2AR-NanoLuc-LOV). As in *Figure 2B–C*, the intracellular NanoLuc fusion construct released the mCherry-Zdk1 from the membrane in response to furimazine. Importantly, additional blue light did not result in further detectable release of mCherry-Zdk1 (*Figure 2—figure supplement 1A–C*). However, the extracellular fusion construct did not release mCherry-Zdk1 from the membrane in response to furimazine. Subsequent blue light delivered to the same cell resulted in the release of mCherry-Zdk1, indicating the LOVTRAP construct was functional (*Figure 2F–H*). To confirm that this difference was not due to higher expression levels of the intracellular NanoLuc versus extracellular NanoLuc fusions, we measured robust NanoLuc bioluminescence emission from both constructs (*Figure 2—figure supplement 1D*). From this experiment, we conclude that Nano-Luc activation of LOV is proximity-dependent, as moving NanoLuc only ~5 nm away from LOV prevents its activation.

## NanoLuc integration into SPARK2

SPARK1 (*Kim et al., 2017*) is an 'AND' logic gate transcription factor, which detects the coincidence of both a PPI and externally delivered blue light (*Figure 1A*). In this tool, protein A is fused to an evolved LOV domain (eLOV) (*Wang et al., 2017*), a protease cleavage sequence (TEVcs: ENLYFQM), and a transcription factor (e.g., Gal4). eLOV's alpha helical C-terminus cages the fused TEVcs in the dark, preventing steric access and reducing background cleavage. The other component of SPARK1 is a low affinity protease (TEVp) fused to protein B. In order for TEVp to cut the TEVcs and release Gal4, a PPI must occur between proteins A and B, *and* blue light must induce a conformational change in eLOV to allow steric access to the TEVcs (*Figure 1B*). Following TEVcs proteolysis, the released TF translocates to the nucleus and drives reporter gene expression.

To address the limitations of SPARK1 as outlined above, we incorporated NanoLuc to control the LOV domain within SPARK. We started with the SPARK1 constructs that have previously been used to detect the PPI between the GPCR beta-2 adrenergic receptor (β2AR) and its cytosolic effector arrestin; this interaction can be stimulated with the β2AR agonist isoetharine. We fused NanoLuc to arrestin-TEVp and co-expressed this construct in HEK293T cells along with β2AR-eLOV-TEVcs-Gal4 and the reporter gene UAS-Citrine (*Figure 3A*). This 'SPARK2' tool gave clear light-dependent and

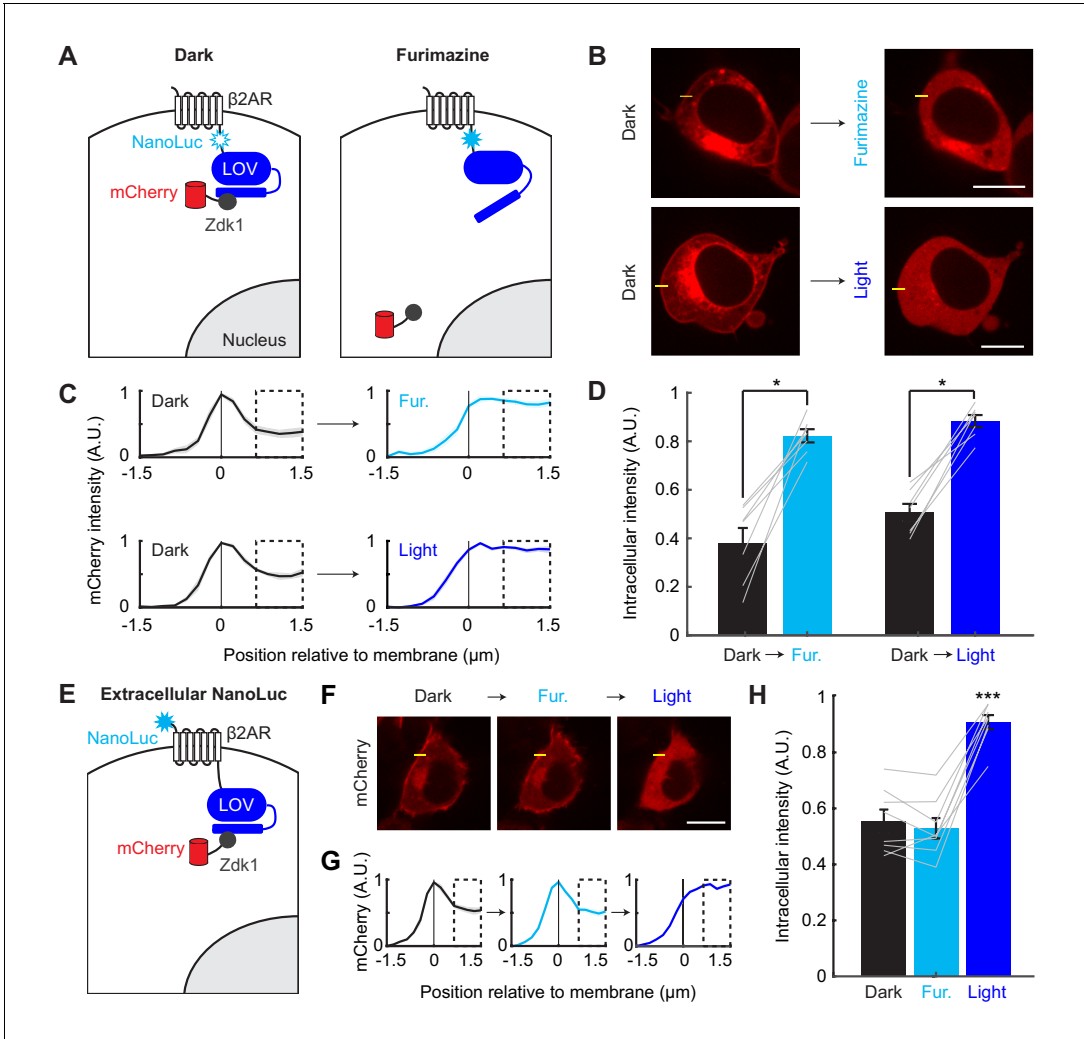

**Figure 2.** Efficient and proximity-dependent NanoLuc-LOV BRET detected using LOVTRAP. (**A**) Schematic of NanoLuc fused to membrane-localized protein β2AR and asLOV2 in LOVTRAP. When asLOV2 is activated by NanoLuc (or blue light), mCherry-Zdk1 dissociates from the C-terminus of asLOV2. The asLOV2 contains a V416L mutation that slows its off kinetics (return to the dark state conformation), and consequently slows the rate of Zdk1 recapture (V416L reset $t_{1/2}$=496 s). (**B**) Example immunofluorescence images of HEK293T cells expressing mCherry-Zdk1 and β2AR-NanoLuc-LOV. Left: mCherry expression is visible along the cell membrane in the dark. Right: following either furimazine for 1 min or blue light for 10 s, mCherry-Zdk1 relocalizes to the cytoplasm. The yellow line indicates pixels quantified for mCherry signal. Scale bars, 11 μm. (**C**) Mean normalized mCherry fluorescence measured across the cell membrane for the same cells before and after furimazine (top), or before and after blue light (bottom; n = 7 cells for both conditions). Dashed box indicates intracellular region quantified in panel **D**). Data plotted as mean ±s.e.m. (**D**) The mean normalized cytosolic mCherry fluorescence increased following either furimazine or light exposure (Dark vs Furimazine: 0.38 ± 0.061 vs 0.82 ± 0.027, n = 7 cells, Wilcoxon's signed-rank test, *p=0.016. Dark vs Light: 0.51 ± 0.035 vs 0.88 ± 0.025, n = 7 cells, Wilcoxon's signed-rank test, *p=0.016). The Furimazine/Dark ratio of cytosolic mCherry fluorescence was comparable to the Light/Dark ratio (Furimazine/Dark: 2.75 ± 0.68, Light/Dark: 1.79 ± 0.12, n = 7 cells each group, Wilcoxon's ranksum test: p=0.84). Data plotted as mean ±s.e.m. (**E**) Schematic of NanoLuc fused to extracellular N-terminus of β2AR-asLOV2 in LOVTRAP. When NanoLuc is expressed extracellularly, we do not expect to observe NanoLuc-asLOV activation with furimazine. As a result, mCherry-Zdk1 should remain at the membrane. (**F**) Example immunofluorescence images of HEK293T cells expressing mCherry-Zdk1 and NanoLuc-β2AR-LOV. Left: mCherry expression is visible along the cell membrane in the dark. Middle: following furimazine for 1 min, mCherry-Zdk1 remains along the cell membrane. Right: following 10 s of blue light, mCherry-Zdk1 relocalizes to the cytosol. The yellow line indicates pixels quantified for mCherry signal. Scale bars, 11 μm. (**G**) Mean normalized mCherry fluorescence measured across the cell membrane for the same cells in the dark, after furimazine, and then after blue light (n = 8 cells). Dashed box indicates intracellular region quantified in panel **H**). Data plotted as mean ±s.e.m. (**H**) With the extracellular NanoLuc, the mean normalized cytosolic mCherry fluorescence did not increase following furimazine. However, subsequent blue light exposure resulted in an increase in the mean normalized cytosolic mCherry fluorescence (Dark: 0.56 ± 0.040; Furimazine: 0.53 ± 0.036; Light: 0.91 ± 0.025; n = 8 cells each group, 1-way ANOVA, $F_{(2,21)}$=37.49, p = 1.18e-7; Tukey's multiple comparison's test, ***p < 0.001 Light compared to all other conditions). Data plotted as mean ± s.e.m. See also *Figure 2—figure supplement 1*.

DOI: https://doi.org/10.7554/eLife.43826.003

*Figure 2 continued on next page*

*Figure 2 continued*

The following source data and figure supplements are available for figure 2:

**Source data 1** Excel spreadsheet containing fluorescence intensity values used to generate panels C-D and G-H.

DOI: https://doi.org/10.7554/eLife.43826.006

**Figure supplement 1.** Quantification and controls for LOVTRAP detection of NanoLuc-LOV activation.

DOI: https://doi.org/10.7554/eLife.43826.004

**Figure supplement 1—source data 1.** Excel spreadsheet containing fluorescence intensity and luminescence values used to generate panels B-D.

DOI: https://doi.org/10.7554/eLife.43826.005

isoetharine-dependent Citrine expression, as we previously observed with SPARK1, showing that introduction of 19 kD NanoLuc does not impair recognition or sterics (*Figure 3—figure supplement 1A–B*). We then tested whether SPARK2, unlike SPARK1, could also be gated by the small-molecule luciferin, furimazine. *Figure 3B* shows that addition of isoetharine and 10 µM furimazine for 15 min (rather than external blue light) results in robust Citrine expression ~8 hr later. Minimal Citrine expression was observed in the absence of furimazine, or with furimazine but no isoetharine.

To test the generality of SPARK2's design, we also incorporated two other GPCR/arrestin pairs that previously showed light- and agonist-dependent transcription in SPARK1 (*Kim et al., 2017*). *Figure 3B* shows that SPARK2 reporters for AVPR2 (arginine vasopressin receptor 2) and DRD1 (dopamine receptor type I) both give furimazine- and agonist-dependent expression of Citrine. Together, these results demonstrate that NanoLuc can indeed regulate the LOV domain of SPARK2 in place of external blue light, enabling users to toggle between luciferin-control and light-control of SPARK2 for PPI detection in living cells.

## Luciferin-gating of SPARK2 produces higher PPI-specificity than light-gating

We hypothesized that the use of NanoLuc for LOV activation in a nested 'AND' gate may also make SPARK2 a more specific tool for PPI detection. In SPARK1 (and in SPARK2 under light control), TEVp recruitment to TEVcs depends on protein A-protein B interaction, but LOV-TEVcs uncaging depends only on exogenous blue light. Thus, during the time period that LOV is open, even in the absence of an A-B interaction, TEVp may interact with TEVcs (if expression levels approach the TEVp-TEVcs $K_m$). This source of background should be eliminated when using luciferin-gated SPARK2, as LOV-TEVcs uncaging is regulated by NanoLuc in a fashion that depends on A-B interaction. In this case, transient interactions between TEVp and TEVcs at high SPARK2 expression levels may not be adequate to produce TEVcs cleavage and TF release.

To test this hypothesis, we repeated the SPARK2 experiment for β2AR/arrestin in HEK293T cells, comparing furimazine- and light-gating by microscopy and single-cell analysis (*Figure 3C* and *Figure 3—figure supplement 1C*). The highest Citrine/HA intensity ratios were observed with external blue light and isoetharine (+light/PPI condition); however, we measured substantial background when light was delivered but isoetharine was omitted (+light/no PPI condition; *Figure 3D*). On the other hand, when we used luciferin (furimazine) to uncage the LOV domain instead of external blue light, we found that background Citrine signal was eliminated (no Citrine expression in the +luciferin/no PPI condition; *Figure 3D*). We could still observe robust Citrine/HA intensity ratios when using luciferin and isoetharine (+luciferin/PPI condition; *Figure 3D*). Anti-HA staining for the protease showed similar expression levels of SPARK2 components across conditions.

When directly compared to SPARK1 with light-gating, SPARK2 with nested luciferase-gating also produced a higher ±PPI signal ratio (*Figure 3—figure supplement 1A–B*). We conclude that luciferin-gating of SPARK2 produces higher PPI-specificity than light-gating, due to the incorporation of a PPI-dependent BRET mechanism for LOV domain control that eliminates PPI-independent background.

## Intermolecular BRET through a nested 'AND' gate is essential for improved specificity

In SPARK2, a nested 'AND' gate is achieved by fusing NanoLuc to the arrestin-TEVp – meaning that first, both the furimazine and the PPI are required for the LOV domain to be uncaged, and then

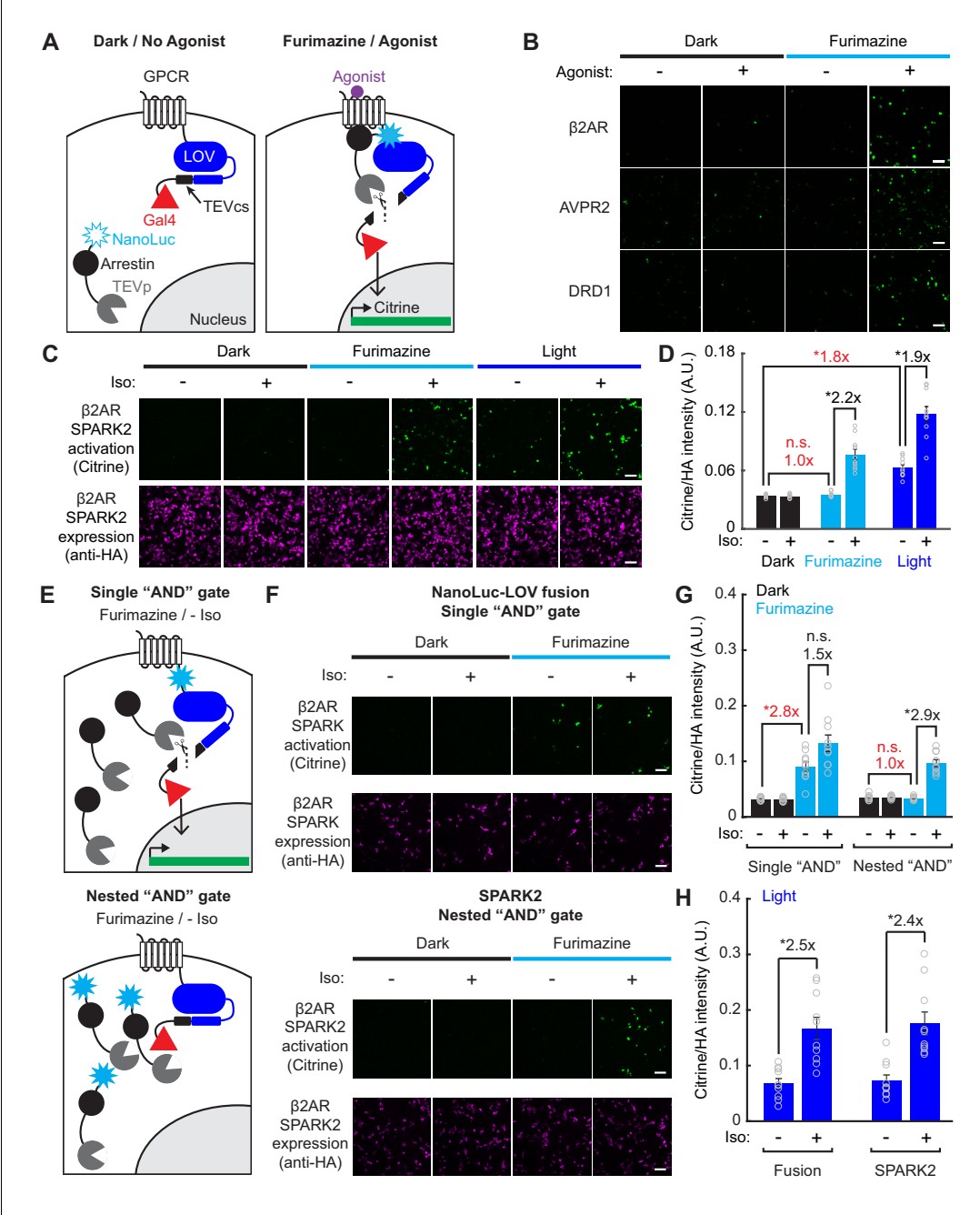

**Figure 3.** Reduced background during transcriptional read-out of PPIs with a nested 'AND' gate in SPARK2. (**A**) Schematic of NanoLuc fused to arrestin-TEVp, co-expressed with GPCR-LOV-TEVcs-Gal4 and UAS-Citrine. In the dark and in the absence of the GPCR's agonist, the TEVcs is caged by the LOV domain and the NanoLuc-arrestin-TEVp is not in proximity to the TEVcs. In the presence of both furimazine and agonist, NanoLuc-arrestin-TEVp is recruited to the GPCR, allowing NanoLuc to uncage eLOV and the TEVp to cleave the TEVcs. The released Gal4 then drives transcription of a UAS reporter gene. (**B**) Example immunofluorescence images of UAS-Citrine expression in HEK293T cells ~ 8 hr following 10 μM furimazine and agonist (20 min 100 μM isoetharine for β2AR, 15 min 10 μM vasopressin for AVPR2, 2 hr 10 μM dopamine for DRD1). Control conditions without agonist and/or without furimazine had low Citrine expression. Scale bars, 100 μm. (**C**) HEK293T cells were transfected with β2AR SPARK2 components as in panel A). SPARK2 activation (Citrine reporter) and SPARK2 expression (HA tag on NanoLuc-arrestin-TEVp) were imaged ~8 hr after 15 min of exposure to 10 μM furimazine or light and 10 μM isoetharine. (**D**) All HA-positive cells were quantified by their Citrine to HA fluorescence intensity ratio. Background fold changes in Furimazine/-Iso and Light/-Iso signal are displayed in red, while ±Ligand signal ratios are displayed in black. There was no additional background signal in the Furimazine/-Iso condition compared to the Dark/-Iso condition (1.0-fold), whereas the Light/-Iso condition had a significant 1.8-fold increase in background compared to the Dark/-Iso condition. There was a 2.2-fold ± Iso signal ratio with furimazine, and a 1.9-fold ± Iso signal ratio with light (n = 10 FOVs each group, two-way ANOVA, interaction $F_{(2,54)}$=25.49, p = 1.61e-8; Tukey's multiple comparison's test, *p < 0.001). Data

*Figure 3 continued*

plotted as mean ± s.e.m. (**E**) Top: Schematic illustrating potential for background TEVcs cleavage in the absence of a PPI using a luciferase-mediated single 'AND' gate (NanoLuc directly fused to N-terminus of LOV domain). Bottom: Schematic illustrating negligible background TEVcs cleavage in the absence of a PPI using the luciferase-mediated nested 'AND' gate in SPARK2 (NanoLuc fused to arrestin-TEVp). (**F**) HEK293T cells were transfected either with a modified β2AR-NanoLuc-LOV direct fusion SPARK construct and Arrestin-TEVp (top) or β2AR SPARK2 components (bottom). SPARK activation (Citrine reporter) and SPARK expression (HA tag on Arrestin-TEVp or NanoLuc-arrestin-TEVp) were imaged ~8 hr after continuous exposure to 10 µM furimazine and 10 µM isoetharine. Scale bars, 100 µm. (**G**) Data were quantified as in panel D. In the modified fusion SPARK (single 'AND' gate), there was not a robust ±Iso signal ratio with furimazine (1.5-fold, not significant, n.s.), and a 2.8-fold background increase due to furimazine alone (n = 10 FOVs each group, two-way ANOVA, interaction $F_{(2,54)}$=14.11, p = 1.18e-5; Tukey's multiple comparison's test, *p < 0.001). In SPARK2 (nested 'AND' gate), there was a 2.9-fold ± Iso signal ratio with furimazine, and no significant background due to furimazine alone (1.0-fold, n.s.; n = 10 FOVs each group, two-way ANOVA, interaction $F_{(2,54)}$=9.82, p = 2.0e-4; Tukey's multiple comparison's test, *p < 0.001). Data plotted as mean ± s.e.m. (**H**) HEK293T cells transfected as in panel F) were exposed to blue light for 5 min, with or without 10 µM isoetharine. The fusion and SPARK2 conditions resulted in similar ±Iso signal ratios with light, 2.5- and 2.4-fold, respectively (Data analyzed in two-way ANOVA with data from panel (G). Fusion: n = 10 FOVs each group, two-way ANOVA, interaction $F_{(2,54)}$=14.11, p = 1.18e-5; SPARK2: n=10 FOVs each group, two-way ANOVA, interaction $F_{(2,54)}$ = 9.82, p = 2.0e-4; Tukey's multiple comparison's test, *p < 0.01). Data plotted as mean ± s.e.m. See also *Figure 3—figure supplement 1*.
DOI: https://doi.org/10.7554/eLife.43826.007

The following source data and figure supplements are available for figure 3:

**Source data 1.** Excel spreadsheet containing fluorescence ratio intensity values used to generate panels D, G, and H.
DOI: https://doi.org/10.7554/eLife.43826.010
**Figure supplement 1.** Direct comparison of SPARK1 versus SPARK2 and raw fluorescence intensities for SPARK2 characterization.
DOI: https://doi.org/10.7554/eLife.43826.008
**Figure supplement 1—source data 1** Excel spreadsheet containing fluorescence intensity ratio values used to generate panel B.
DOI: https://doi.org/10.7554/eLife.43826.009

both the LOV domain uncaging and the PPI are required for the TEVcs cleavage. If NanoLuc were installed in SPARK2 directly adjacent to LOV rather than on the TEVp construct, then we would not expect an improvement in specificity in the +furimazine/-PPI condition compared to the intermolecular BRET design of SPARK 2 (*Figure 3E*). Intramolecular furimazine uncaging of LOV should still occur, but as it now only requires the presence of luciferin but not A-B interaction, the TF activation would no longer have a 'nested' dependence on the PPI. We tested this concept by generating a modified β2AR-NanoLuc-LOV-TEVcs-Gal4 fusion construct, which when co-expressed with arrestin-TEVp, employs only a single 'AND' gate relying on furimazine and a PPI (*Figure 3E*). Compared to the nested 'AND' gate design of SPARK2, the single 'AND' gate design resulted in high background in the +furimazine/-PPI condition, comparable to the background observed with +light/-PPI (*Figure 3F–H*). Importantly with 5 min of blue light, the NanoLuc-LOV fusion construct produced a similar ±PPI signal ratio as light-gating with SPARK2, suggesting that there is no steric hindrance due to the fusion of NanoLuc to LOV (*Figure 3H*). These data demonstrate that the nested 'AND' gate via intramolecular BRET is essential for driving the improved specificity in PPI detection with SPARK2.

## SPARK2 is compatible with an orthogonal luciferase as the transcriptional readout for high-throughput drug screens

In addition to its utility for BRET, luciferase is also a convenient and scalable reporter commonly used for transcriptional assays, which we employed in our SPARK1 study (*Kim et al., 2017*). To check if we could continue to use luciferase as the readout, while using the luciferase NanoLuc within SPARK2, we transfected HEK293T cells with β2AR SPARK2 components in addition to the reporter UAS-Firefly luciferase (*Photinus pyralis*; FLuc). FLuc uses an orthogonal substrate to NanoLuc (Beetle luciferin) and also luminesces at a different wavelength (535 nm instead of 450 nm (*Germain-Genevois et al., 2016*); *Figure 4A*). We performed a SPARK2 assay as before, but this time measuring FLuc reporter luminescence instead of UAS-Citrine fluorescence following furimazine and agonist exposure. We observed 11.6-fold higher FLuc luminescence in cells exposed to furimazine and isoetharine for 20 min compared to cells exposed to only furimazine (*Figure 4B*). As there was no FLuc luminescence detected in the control conditions, this confirms that we are detecting only the FLuc luminescence and not any cross-over NanoLuc luminescence.

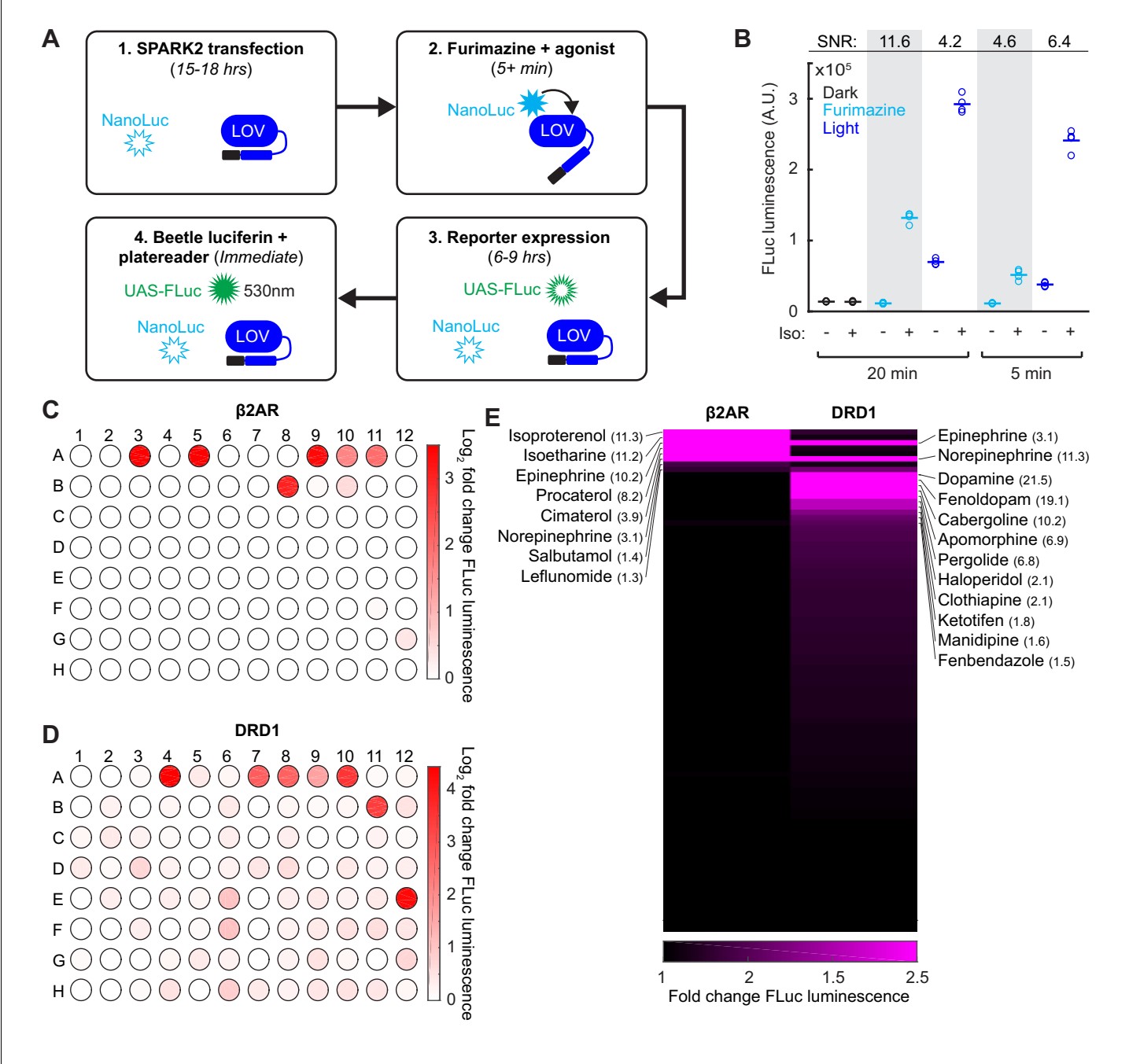

**Figure 4.** SPARK2 is compatible with an orthogonal luciferase reporter to enable high-throughput drug-screens for GPCR activation. (**A**) Schematic of the general timeline for SPARK2 assays using an orthogonal luciferase reporter such as FLuc. For simplicity, only the luciferases and LOV domain are illustrated (unfilled luciferases indicate no luminescence; filled luciferases indicate luciferin substrate-mediated luminescence). (**B**) HEK293T were transfected with β2AR SPARK2 components and UAS-FLuc.~8 hr after SPARK2 labeling with 10 μM furimazine/blue light and 10 μM isoetharine, the UAS-FLuc luminescence recorded using a platereader (n = 4 replicates each condition). The highest ±Iso signal ratio was achieved using 20 min exposure to furimazine (11.6-fold). 5 min of blue light resulted in a ± Iso signal ratio of 6.4-fold, whereas 20 min of blue light or 5 min of furimazine resulted in similar ±Iso signal ratios of 4.2- and 4.6-fold, respectively. (**C–D**) HEK293T cells were transfected with either β2AR or DRD1 SPARK2 components and UAS-FLuc. Cells were exposed to 10 μM furimazine and 5 μM each of 92 different drugs for 1 hr (BioMol FDA-approved compound library). In addition, we performed two DMSO vehicle measurements (**A1–A2**), and two positive control spike-ins (isoetharine in A3 and dopamine in A4). ~8 hr later, FLuc luminescence was measured on a platereader. The Log$_2$ fold change in FLuc luminescence is plotted corresponding to each compound's position in the 96-well plate. Data plotted as the average of two biological replicates. Compound list and positions in *Figure 4—source data 2*. (**E**) Sorted heatmap of fold changes in FLuc luminescence for β2AR and DRD1 SPARK2 for all 94 compounds (excluding DMSO replicates).

*Figure 4 continued on next page*

*Figure 4 continued*

Example compound hits for each receptor are labeled. Heatmap thresholded from 1 to 2.5 for visualization, mean fold-changes for selected hits listed in parentheses. Fold-changes for all compounds listed in *Figure 4—source data 2*.

DOI: https://doi.org/10.7554/eLife.43826.011

The following source data and figure supplements are available for figure 4:

**Source data 1.** Excel spreadsheet containing luminescence values used to generate panel B.

DOI: https://doi.org/10.7554/eLife.43826.014

**Source data 2.** Excel spreadsheet containing SPARK2 drug screen compounds and mean fold-changes (n = 2 biological replicates) to generate panels C-E.

DOI: https://doi.org/10.7554/eLife.43826.015

**Figure supplement 1.** Quantification and controls for proximity-dependence of SPARK2.

DOI: https://doi.org/10.7554/eLife.43826.012

**Figure supplement 1—source data 1.** Excel spreadsheet containing luminescence values used to generate panels B-D.

DOI: https://doi.org/10.7554/eLife.43826.013

We then used this multiwell plate assay configuration to again compare the specificity of SPARK2 under light versus furimazine-gating. *Figure 4B* shows that when SPARK2 is transiently expressed in HEK293T cells, FLuc luminescence background is significant in the +light/no PPI condition but undetectable in the +furimazine/no PPI condition (both 5 min and 20 min treatments). The consequence is that the highest ±PPI signal ratio is obtained with luciferin gating for 20 min (*Figure 4B*). At 5 min, the reduction in -PPI background with luciferin is offset by reduced +PPI signal, resulting in a lower ±PPI signal ratio compared to light-gating. Overall, these results, and the single cell imaging results above, support the notion that nested luciferin-gated SPARK2 is a lower-background, higher-specificity PPI reporter alternative to SPARK1/2 with light-gating.

Given the higher specificity of SPARK2, and the ability to perform an all-drug-gated assay, we then implemented a high-throughput compound sub-library screen to test for β2AR or DRD1 receptor activation (*Figure 4C–D*). In this configuration, rather than delivering timed blue light, we simply added the additional furimazine substrate along with the compounds, which makes our tool compatible with robotics-based drug-screening platforms if desired. We observed robust FLuc reporter activation for our positive control spike-in compounds isoetharine and dopamine, both delivered at 5 µM. For β2AR, we detected several compounds with >3 fold increases in FLuc reporter expression relative to a DMSO vehicle control, including known agonists such as isoproterenol, epinephrine, procaterol, cimaterol, and norepinephrine (*Figure 4E* and *Figure 4—source data 2*). We could also detect weaker agonists such as the known partial β2AR agonist salbutamol and leflunomide, which drove 1.4- and 1.3-fold increases in FLuc reporter expression, respectively. Interestingly, dopamine, which is a known β2AR partial agonist, did not drive an increase in FLuc expression, supporting the notion that dopamine may act as a G-protein-biased β2AR partial agonist (*Kobilka and Deupi, 2007*; *Swaminath et al., 2004*).

For DRD1, we detected many more compounds that drove >1.5 fold increases in FLuc reporter expression. Notably, known partial and full agonists such as fenoldopam, cabergoline, apomorphine, and pergolide drove >6 fold increases in FLuc reporter expression (*Figure 4E* and *Figure 4—source data 2*). Catecholamine β2AR agonists epinephrine and norepinephrine also drove a 3- and 11-fold increase in FLuc expression for DRD1 activation. Surprisingly, the anti-psychotic haloperidol, which is a known DRD2 antagonist (*Leysen et al., 1992*), drove a two-fold increase in FLuc expression for DRD1 activation. These screening results highlight the utility of SPARK2 for the sensitive and high-throughput detection of GPCR activation.

## Proximity-dependence of NanoLuc-LOV interaction using SPARK2

Although we demonstrated using the LOVTRAP assay above that luciferase-LOV BRET is proximity-dependent, we examined proximity-dependence in the context of SPARK2 as well. To do so, we compared the β2AR-arrestin SPARK2 constructs from *Figure 4A* to control constructs in which Nano-Luc is instead targeted to the cell membrane, where it is in the vicinity of but not directly interacting with the LOV domain (*Figure 4—figure supplement 1A*). Whereas β2AR-arrestin SPARK2 gives a 9.0-fold signal increase when agonist is added versus withheld (both in the presence of furimazine), the control condition gave much lower reporter gene expression (*Figure 4—figure supplement 1B*).

Confirming that this difference was not due to higher expression levels of the NanoLuc-arrestin-TEVp compared to the membrane-bound NanoLuc, we measured robust NanoLuc bioluminescence emission from both constructs (*Figure 4—figure supplement 1C*). In addition, we delivered blue light instead of furimazine and measured comparable levels of FLuc reporter signal in response to agonist for both conditions, suggesting that the other SPARK2 components were similarly expressed as well, and functional (*Figure 4—figure supplement 1D*). These results again support the notion that NanoLuc must be in close proximity (<5 nm) to the LOV domain in order to efficiently mediate its opening.

## Extending NanoLuc BRET to other molecular actuators to detect trans-cellular interactions with SPARK2

We wondered whether the SPARK2 platform could be extended to detect trans-cellular interactions and not only intracellular PPIs. To do so, we envisioned making two modifications to SPARK2. First, for protein A, instead of a regular GPCR, we would install a light-sensitive GPCR chimera, such as 'oβ2AR', previously used to enable light induction of G-protein signaling in neurons in vivo (*Airan et al., 2009*). Second, to activate oβ2AR across cell-cell junctions, we would utilize a second copy of the luciferase NanoLuc on the 'sender' cells. With this design, addition of furimazine to cells would be expected to (1) turn on oβ2AR *selectively* at cell-cell junctions that are positive for Nano-Luc sender and oβ2AR receiver, and (2) enable NanoLuc-arrestin-TEVp recruited to activated oβ2AR to open the LOV domain and permit cleavage and release of the TF. Reporter gene expression should then be observed specifically in 'receiver' cells in contact with NanoLuc-expressing 'sender' cells.

For this system to work, oβ2AR (a chimera between bovine rhodopsin and β2AR (*Airan et al., 2009*)) must recruit NanoLuc-arrestin-TEVp upon activation, and NanoLuc must be able to activate oβ2AR in place of blue light. While *Gaussia* luciferase has previously been fused to the N-terminus of channelrhodopsins (*Berglund et al., 2013*; *Berglund et al., 2016*; *Park et al., 2017*) to regulate their activity in vivo ('luminopsins'), luciferases have not previously been shown to turn on light-sensitive GPCR chimeras such as oβ2AR. Furthermore, proximity-dependent intermolecular luciferase regulation of retinal-binding proteins has also not been previously demonstrated.

To explore this extension of SPARK2, we started by creating an oβ2AR SPARK2 construct with NanoLuc fused directly to its N-terminus. HEK293T cells expressing this construct along with Nano-Luc-arrestin-TEVp and UAS-Citrine (*Figure 5A*) were incubated with the co-factor 9-cis-retinal (9cr), as HEK293T do not naturally produce 9cr. As a positive control, we delivered 15 min of blue light to both uncage the eLOV domain and activate oβ2AR. We observed significantly more Citrine-positive cells in the +light/+9cr condition than in the +light/−9cr condition, indicating that the oβ2AR is functional (*Figure 5—figure supplement 1A–B*). In a separate experimental condition, we delivered 30 min of furimazine and 9cr and observed significantly more Citrine-positive cells compared to control conditions (*Figure 5B–C*). These results suggest that both extracellular NanoLuc-oβ2AR BRET and intracellular NanoLuc-LOV BRET are occurring in the same cells, enabling TF release to produce reporter signal.

Next, we moved to test *inter*cellular activation of oβ2AR by NanoLuc. Instead of fusing NanoLuc directly to the N-terminus of oβ2AR, we expressed NanoLuc on 'Sender' cells, and oβ2AR SPARK2 components separately in 'Receiver' cells, as shown in *Figure 5D*. For display of NanoLuc on senders, we used the pre-synaptic adhesion protein neurexin as a scaffold, and the extracellular domain of ICAM-1 (*Talay et al., 2017*) to extend NanoLuc further out from the cell surface. After separately transfecting the two HEK293T populations, we trypsinized the cells to lift them, and re-plated them together in the same dish at a 10:1 sender: receiver ratio (*Figure 5E*).

We exposed the co-plated cells to furimazine and 9cr for 20 min and observed Citrine-positive cells ~ 8 hr later. To positively identify Sender and Receiver cells, we immunostained for HA on the Neurexin-NanoLuc Sender construct and for Flag on the oβ2AR Receiver construct (*Figure 5F*). To quantify specificity, we counted the fraction of Citrine-positive Receiver cells that were contacting HA-positive Sender cells (87%; n = 43 cells from 26 FOVs). It is possible that the few Citrine + cells not in contact with NanoLuc +Senders were previously in contact, but the Senders were dislodged during the 8 hr incubation and washing period. To quantify Citrine expression in all Receiver cells, we calculated the ratio of Citrine fluorescence to Flag immunostain detecting the oβ2AR component. We observed a higher overall Citrine/Flag intensity ratio in the condition with furimazine and

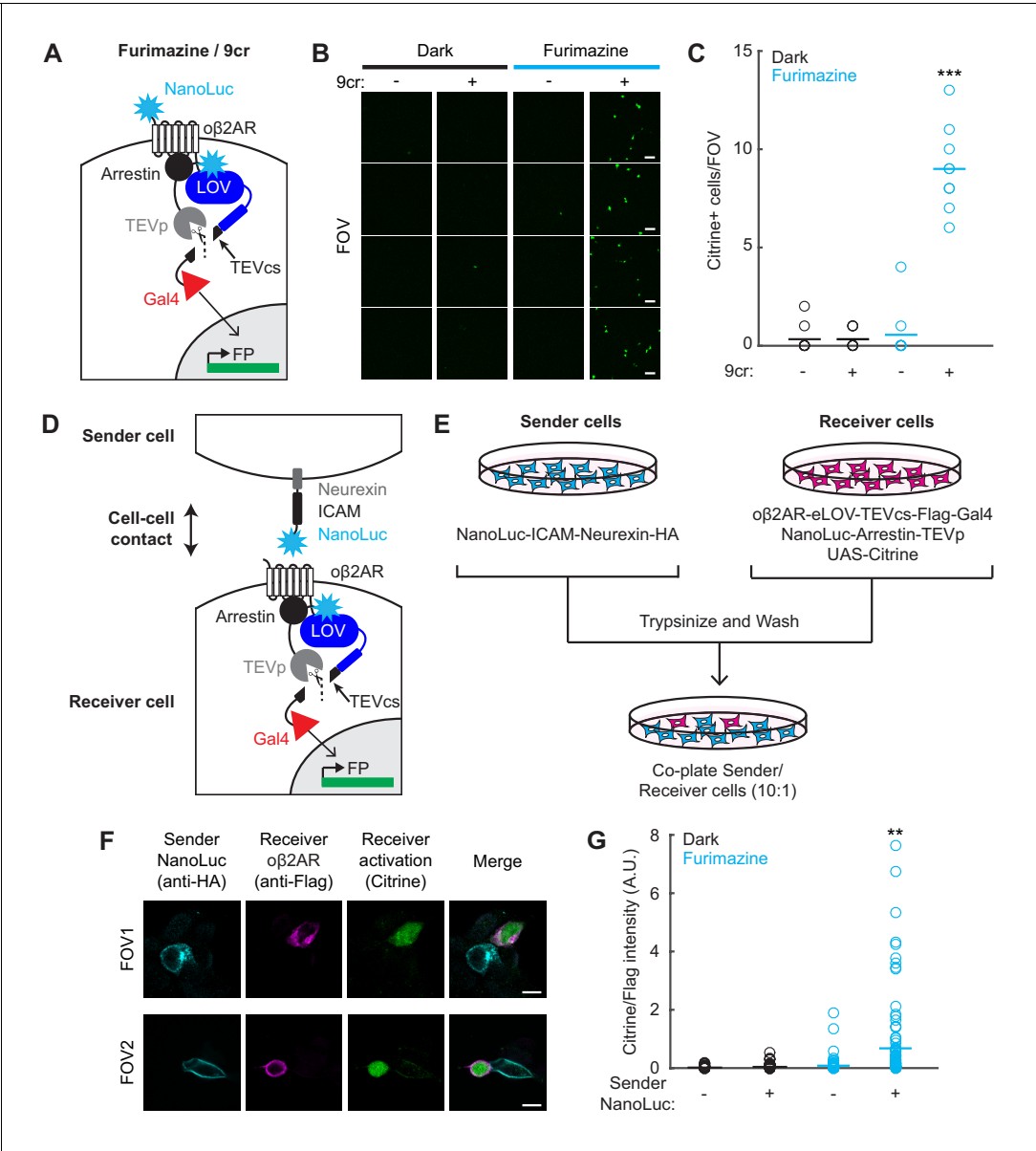

**Figure 5.** Detection of cell-cell contacts using NanoLuc-oβ2AR activation with SPARK2. (**A**) Schematic of an extracellular NanoLuc fused to oβ2AR-eLOV-TEVcs-Gal4, co-expressed with NanoLuc-arrestin-TEVp and UAS-Citrine. (**B**) Citrine expression in HEK293T cells infected with lentiviruses expressing components in A) and exposed to 10 μM furimazine and 50 μM 9-cis-retinal (9cr) for 30 min. In control conditions, 9cr was omitted and/or furimazine was omitted. Scale bar, 100 μm. (**C**) There were significantly more Citrine-positive cells per field of view ~24 hr following furimazine and 9cr exposure compared to in control conditions (Dark/−9cr: 0.33 ± 0.24; Dark/+9cr: 0.33 ± 0.17; Fur/−9cr: 0.56 ± 0.44; Fur/+9cr: 9.0 ± 0.71; n = 9 fields of view each condition; two-way ANOVA, interaction $F_{(1,32)}$=91.32, p = 6.80e-11; Tukey's multiple comparison's test ***p < 0.001 compared to all other conditions). (**D**) Schematic for detecting cell-cell contacts in which a Sender cell expresses NanoLuc-ICAM-Neurexin, and a Receiver cell expresses oβ2AR-eLOV-TEVcs-Gal4, NanoLuc-Arrestin-TEVp, and UAS-Citrine. (**E**) Experimental paradigm for co-plating Sender and Receiver cells in a 10:1 ratio. (**F**) Example immunofluorescence images of Citrine-positive Receiver cells that are adjacent to HA-positive Sender cells expressing NanoLuc. The Flag stains against Receiver cells expressing the oβ2AR SPARK2 components. 87% of all detected Citrine-positive Receiver cells were adjacent to an HA-positive Sender cell. Scale bars, 10 μm. (**G**) Quantification of Citrine/Flag fluorescence intensity ratios for all Flag-positive cells. The Citrine/Flag intensity ratio was significantly higher following furimazine exposure when Sender NanoLuc was expressed compared to in control conditions where NanoLuc was not expressed in the Sender cells (Dark/Sender -NanoLuc: 0.016 ± 0.0037, n = 92 cells; Dark/Sender +NanoLuc: 0.039 ± 0.0074, n = 109 cells; Fur/Sender -NanoLuc: 0.082 ± 0.025, n = 95 cells; Fur/Sender +NanoLuc: 0.68 ± 0.14, n = 104 cells; two-way ANOVA, interaction $F_{(1,396)}$=15.6, p = 9.28e-5; Tukey's multiple comparison's test, **p < 0.001 compared to all other conditions). See also *Figure 5—figure supplement 1*.
DOI: https://doi.org/10.7554/eLife.43826.016

The following source data and figure supplements are available for figure 5:

*Figure 5 continued on next page*

*Figure 5 continued*

**Source data 1.** Excel spreadsheet containing cell count and fluorescence intensity ratio values used to generate panels C and G.
DOI: https://doi.org/10.7554/eLife.43826.019
**Figure supplement 1.** Quantification and controls for detection of cell-cell contacts using NanoLuc-oβ2AR activation with SPARK2.
DOI: https://doi.org/10.7554/eLife.43826.017
**Figure supplement 1—source data 1.** Excel spreadsheet containing cell counts and fluorescence intensity ratio values used to generate panels B and D-E.
DOI: https://doi.org/10.7554/eLife.43826.018

Sender NanoLuc compared to control conditions with furimazine omitted or Sender NanoLuc omitted (*Figure 5G* and *Figure 5—figure supplement 1C*). We confirmed that in the absence of 9cr, there was no significant increase in the Citrine/Flag intensity with either blue light or furimazine (*Figure 5—figure supplement 1D–E*). These results provide a proof-of-concept demonstration that SPARK2 can be combined with intercellular NanoLuc-oβ2AR BRET to enable luciferin-gated, bioluminescence-mediated transcriptional readout of cell-cell contacts.

## Discussion

Motivated by specificity and versatility limitations of our previously reported SPARK1 PPI detection tool, here we developed improved, second-generation SPARK2. SPARK2 employs a novel design not previously used for PPI detection. Via a single modification of SPARK1—addition of a luciferase moiety onto the protease component of SPARK—SPARK2 gains two new capabilities: the ability to toggle between light-gating and luciferin-gating for temporal specificity, and greatly reduced background leading to a more PPI-specific tool. We showed that SPARK2 could be used to detect transient arrestin recruitment to several activated GPCRs, giving stable and versatile (Citrine, FLuc) reporter gene readout many hours later. Using luciferin-gating, we used SPARK2 for high-throughput compound screening against activation of two GPCRs. We also extended the SPARK2 design to detection of cell-cell contacts, employing trans-cellular luciferase activation of a GPCR-rhodopsin chimera.

At the heart of SPARK2 is a proximity-dependent interaction between the blue light-emitting luciferase NanoLuc and the blue-light absorbing photosensory domain LOV. Luciferases have been widely used for bioluminescence imaging in vivo, free from the constraint of excitation light delivery (*Maguire et al., 2013*; *Tung et al., 2016*). They have also been combined with fluorescent proteins for BRET-based shifting of emission wavelength (*Chu et al., 2016*), real-time detection of cellular PPIs (*Hamdan et al., 2006*; *Pfleger et al., 2006*), and calcium sensing (*Qian et al., 2018*). Recent studies have expanded the range of luciferase capabilities by fusing them to channelrhodopsins (*Berglund et al., 2016*; *Park et al., 2017*) and miniSOG (*Proshkina et al., 2018*), to confer luciferin-control of membrane depolarization and $^1O_2$ generation, respectively. Here, we continue the trend by showing that luciferase can also regulate the activity of LOV domains. We further demonstrate that such regulation is proximity-dependent, both in the context of LOVTRAP and SPARK2. LOV domains form the basis of a wide-range of optogenetic tools, including LOVTRAP (*Wang et al., 2016*), TULIPs (*Strickland et al., 2012*), iLID (*Guntas et al., 2015*), miniSOG (*Shu et al., 2011*), BLINK (*Alberio et al., 2018*), and Magnets (*Kawano et al., 2015*). It seems likely that if these tools were to also incorporate NanoLuc, this would enable users to toggle between light control and luciferin control, as we do for SPARK2. Furthermore, because we have shown that luciferase-LOV regulation is proximity-dependent, one could envision tool variants in which LOV-mediated functions (PPI induction, protein inactivation/activation, channel opening, etc.) are conditional upon PPIs or subcellular localization events designed by the user. Finally, we note that luciferase regulation is likely to extend beyond LOVs. Cryptochromes (e.g. CRY-CIBN (*Kennedy et al., 2010*)) also contain blue-light excitable flavin cofactors, and we showed here that luciferase can also regulate oβ2AR, a retinal-containing blue-light activated GPCR-rhodopsin chimera.

Besides enabling SPARK2 toggling between light-gating and luciferin-gating, incorporation of NanoLuc onto the protease component of SPARK substantially reduced the PPI-independent background of the system. We rationalize this by the logic diagram in *Figure 1D*, showing that both inputs into the committed step (TEVcs cleavage) – LOV opening and TEVp recruitment – require

protein A-protein B interaction. This contrasts with the simpler logic diagram of SPARK1 (or in a version of SPARK where NanoLuc is directly fused to LOV), in which only TEVp recruitment depends on protein A-protein B interaction. A 'double filter' for the analyte of interest (in this case, a PPI) appears to improve the specificity of a reporter for that analyte. We envision that the 'double filter' or nested 'AND' gate concept could also be incorporated into a range of other reporter designs, particularly those with limited dynamic range. Interestingly, there are numerous examples of natural pathways that incorporate 'double filters' to increase specificity, such as tRNA synthetases that catalyze amino acid loading onto cognate tRNAs (*Fersht and Dingwall, 1979*).

Finally, our trans-cellular SPARK2 demonstration adds to the wide array of existing methods for probing cell-cell contacts, including mGRASP (*Kim et al., 2011b*; *Feinberg et al., 2008*), sHRP (*Martell et al., 2016*), LIPSTIC (*Pasqual et al., 2018*), synNotch (*Morsut et al., 2016*), and trans-TANGO (*Talay et al., 2017*). While mGRASP, sHRP, and LIPSTIC enable highly sensitive detection, these tools generate signal in the intercellular space between contacting cell populations, and do not highlight or provide genetic access to the cells themselves, preventing further imaging, selection, characterization by RNA-seq, etc. synNotch and trans-TANGO, by contrast, activate transcription factors in contacting cell populations and thus provide more versatile readouts, but they lack temporal-gating which can contribute to high background, and they could potentially perturb normal physiology by introducing non-natural intercellular protein-protein interactions. Trans-cellular SPARK2 has complementary features to these existing methods, although further optimization and characterization are required to make it a broadly useful tool for the study of trans-cellular interactions.

# Materials and methods

## Key resources table

| Reagent type (species) or resource | Designation | Source or reference | Identifiers | Additional information |
|---|---|---|---|---|
| Cell line (HEK293T) | HEK293T | ATCC | | Tested negative for mycoplasma. |
| Antibody | Rabbit anti-HA | Cell Signaling Technology | C29F4 | 1:100 |
| Antibody | Goat anti-Rabbit AlexaFluor 647 | Life Technologies | A27040 | 1:1000 |
| Antibody | Mouse anti-Flag | Sigma | F3165 | 1:300 |
| Antibody | Goat anti-Mouse AlexaFluor 568 | Life Technologies | A11004 | 1:1000 |
| Recombinant DNA reagent | | | | See *Table 1* for all plasmids used or generated in this study. |
| Commercial assay or kit | Nano-Glo Live Cell Assay System | Promega | | 1:100 substrate |
| Commercial assay or kit | Bright-Glo Luciferase Assay System | Promega | | 1:1 substrate |
| Software, algorithm | Fiji | *Schindelin et al., 2012* | | |
| Software, algorithm | CellSegm toolbox | *Hodneland et al., 2013* | | |
| Software, algorithm | MATLAB R2017a | Mathworks | | |

## Plasmid constructs and cloning

Constructs used for transient expression in HEK293T cells were cloned into the pAAV viral vector, pC3 vector, or pLX208 vector. *Table 1* lists all 18 plasmids used in this study. NanoLuc was amplified

from a construct provided as a gift from Ute Hochgeschwender (Central Michigan University). For LOVTRAP experiments, AsLOV2(V416L) and Zdk1 were amplified from Addgene plasmids #81036 and #81010 (Klaus Hahn), and β2AR was amplified from Addgene plasmid #104841 (Alice Ting). The oβ2AR construct was a gift from Karl Deisseroth (Stanford University). For all constructs, standard cloning procedures were used. PCR fragments were amplified using Q5 polymerase (NEB), and vectors were double-digested with NEB restriction enzymes and ligated to gel-purified PCR products using Gibson assembly. Ligated plasmids were introduced into competent XL1-Blue bacteria via heat shock transformation.

## HEK293T cell culture and transfection

HEK293T cells were obtained from ATCC (tested negative for mycoplasma) and cultured as monolayers in complete growth media: Dulbecco's Modified Eagle Medium (DMEM, Corning) supplemented with 10% Fetal Bovine Serum (FBS, VWR) and 1% (v/v) Penicillin-Streptomycin (Corning, 5000 units/mL of penicillin and 5000 μg/mL streptomycin), at 37°C under 5% $CO_2$. For experimental assays, cells were grown in 6-well, 24-well, or 48-well plates (with or without 7 × 7 mm glass coverslips) pretreated with 50 μg/mL human fibronectin (Millipore).

DNA transfections were performed using either polyethyleneimine (PEI, Polysciences catalog no. 24765; polyethyleneimine HCl Max in $H_2O$, pH 7.3, 1 mg/mL) or Lipofectamine 2000 Transfection Reagent (Invitrogen). Complete transfection protocols are described for each experiment below.

## Immunofluorescence imaging

Confocal imaging was performed with a Zeiss AxioObserver inverted microscope with 10x and 20x air objectives, and a 63x oil-immersion objective. The following combinations of laser excitation and emission filters were used for various fluorophores: Citrine (491 nm laser excitation; 528/38 nm emission), mCherry/Alexa Fluor 568 (561 nm laser excitation; 617/73 nm emission), Alexa Fluor 647 (647 nm excitation; 680/30 nm emission). Acquisition times ranged from 100 to 500 ms. All images were collected with SlideBook (Intelligent Imaging Innovations) and processed with Fiji (*Schindelin et al., 2012*).

## Sample-size estimation and replication

No statistical methods were used to determine sample size, and instead relied on guidelines from previously published works. For immunofluorescence assays, we used 5–10 fields of views for analysis for each condition (technical replicates). For luminescence assays, we used at least four technical replicates. For drug screens, we used two biological replicates. Sample sizes are listed in figure legends. All experiments were replicated at least once (biological replicates). Replicates are listed in figure legends.

## LOVTRAP

HEK293T cells plated on coverslips in 48-well plates were transfected with 1) 35 ng of either P1. pAAV-CMV-β2AR-NanoLuc-AsLOV2(V416L) or P2.pAAV-CMV-NanoLuc-β2AR-AsLOV2(V416L), and 2) 5 ng of P3.pAAV-CMV-mCherry-Zdk1. Per each well, plasmid DNA was mixed with 0.8 μL of PEI max in 10 μL of serum-free DMEM and incubated at room temperature for 20 min. 100 μL of DMEM + 10% FBS was then added to the mixture, and the entire volume was added to a well of the 48-well plate. Following ~15 hr of incubation, coverslips were placed in a 35 mm imaging dish (Corning) and submerged in 100 μL of DPBS. mCherry expression was first imaged on the confocal at 63x magnification in the dark to obtain a baseline image. Cells were then exposed to either 10 s of blue light (491 nm laser from the confocal), or a mixture of 1.25 μL furimazine in 23.75 μL DPBS (1:100) was added to the cells and allowed to diffuse for 1 min. A second image of mCherry expression in the same field of view was then taken. Images were quantified using Fiji and plotted/analyzed in MATLAB (Mathworks).

## SPARK
### Immunofluorescence experiments

For immunofluorescence SPARK2 experiments, HEK293T cells plated directly on 48-well plates were transfected with 1) 35 ng of either P4.pAAV-CMV-DRD1-eLOV-TEVcs-Gal4, P5.pAAV-CMV-AVPR2-

eLOV-TEVcs-Gal4, or P6.pAAV-CMV-β2AR-eLOV-TEVcs-Gal4, 2) 10 ng of P7.pAAV-NanoLuc-β arrestin2-TEVp, and 3) 15 ng of P8.pAAV-UAS-Citrine. For modified NanoLuc-LOV fusion SPARK experiments, HEK293T cells were transfected with 1) 35 ng of P18.pAAV-CMV-β2AR-NanoLuc-eLOV-TEVcs-Gal4, 2) 10 ng of P9.pAAV-CMV-βarrestin2-TEVp, and 3) 15 ng of P8.pAAV-UAS-Citrine. For SPARK1 experiments, HEK293T cells were transfected with 1) 35 ng of P6.pAAV-CMV-β2AR-eLOV-TEVcs-Gal4, 2) 10 ng of P9.pAAV-CMV-βarrestin2-TEVp, and 3) 15 ng of P8.pAAV-UAS-Citrine. Per each well, plasmid DNA was mixed with 0.8 µL of PEI max in 10 µL of serum-free DMEM and incubated at room temperature for 20 min. 100 µL of DMEM +10% FBS was then added to the mixture, and the entire volume was added to a well of the 48-well plate. Following ~15 hr of incubation, the media in each well was replaced with 100 µL of DMEM +10% FBS +20 mM HEPES with or without 10 µM agonist: dopamine for DRD1, vasopressin for AVPR2, or isoetharine for β2AR. For furimazine exposure, a mixture of 1.25 µL furimazine in 23.75 µL of DMEM +20 mM HEPES (1:100) was added to each well. Cells were incubated for 15 min at room temperature in the dark wrapped in aluminum foil (for the initial DRD1 test, cells were incubated for 2 hr). For the light condition, cells were exposed to an LED light array directed over the entire plate (467 nm, 60 mW/cm$^2$, 10–33% duty cycle, 2 s of light every 4 s) for 15 min at 37°C. Following furimazine or light exposure, the media was then removed and replaced with 200 µL DMEM +10% FBS, and the cells were returned to the 37°C incubator for ~8 hr wrapped in aluminum foil. Cells were fixed in 4% paraformaldehyde at room temperature for 15 min, then washed 2x with DPBS. Cells were permeabilized with methanol at −20°C for 10 min, and then washed 2x with DPBS. Cells were stained with primary antibody (1:100 Rabbit anti-HA, Cell Signaling Technology C29F4), diluted in 1% bovine serum albumin blocking solution, at room temperature for 2 hr on a shaker. Cells were stained with secondary antibody (1:1000 Goat anti-Rabbit AlexaFluor 647, Life Technologies A27040), diluted in 1% blocking solution, for 30 min and then washed with DPBS 3x prior to imaging. Citrine and AlexaFluor 647 images were acquired at 10x magnification on the confocal and quantified using the CellSegm toolbox (*Hodneland et al., 2013*) in MATLAB.

## FLuc NanoLuc-SPARK experiments

For NanoLuc-mediated SPARK experiments with an FLuc reporter, HEK293T cells plated directly on six-well plates were transfected with 1) 350 ng of P6.pAAV-CMV-β2AR-eLOV-TEVcs-Gal4, 2) 100 ng of P7. pAAV-CMV-NanoLuc-βarrestin2-TEVp or 100 ng of P9. pAAV-CMV-βarrestin2-TEVp, and 3) 150 ng of P10.pAAV-UAS-FLuc. Per each well, plasmid DNA was mixed with 8 µL of Lipofectamine in 100 µL of serum-free DMEM and incubated at room temperature for 20 min. Media from the well was aspirated and replaced with the DNA-Lipofectamine mixture in 2 mL of serum-free DMEM. Cells were incubated for 3 hr in the 37°C incubator. Cells were then lifted using 300 µL Trypsin (Corning), resuspended in 1.7 mL DMEM +10% FBS, and 100 µL of the cell suspension was re-plated into each well of a 96-well white microplate (Corning). Plates were wrapped in aluminum foil and kept in the 37°C incubator for an additional ~12 hr.

The media was then aspirated and replaced with 100 µL DMEM +10% FBS with or without 10 µM isoetharine. For the furimazine condition, 1.25 µL furimazine in 23.75 µL DMEM +10% FBS +20 mM HEPES was added to each well (1:100), and the plate was kept at room temperature. For the light condition, cells were exposed to an LED light array (467 nm, 60 mW/cm$^2$, 10–33% duty cycle, 2 s of light every 4 s) at 37°C. After stimulation, the media was replaced with 100 µL DMEM +10% FBS, and the plates were wrapped in aluminum foil and incubated in the 37°C incubator for 9 hr prior to luminescence measurements on a plate reader (Tecan Infinite M1000 Pro).

For UAS-FLuc reporter measurements, the Bright-Glo Luciferase Assay System was used (Promega). The Bright-Glo reagent was thawed at room temperature in a water bath prior to usage. Media was aspirated from each well, and 50 µL of DPBS and 50 µL of Bright-Glo reagent were added to each well. Luminescence was immediately analyzed at 25°C using a 1000 ms acquisition time, the Green-1 filter, and linear shaking for 3 s.

For complexed versus nearby assays, HEK293T cells in six-well plates were transfected and re-plated as described above with 1) 350 ng of P6.pAAV-CMV-β2AR-eLOV-TEVcs-Gal4, 2) 100 ng of P7.pAAV-CMV-NanoLuc-βarrestin2-TEVp (complexed) or 100 ng each of both P11.pAAV-CMV-CD4-CIBN-NanoLuc and P9.pAAV-CMV-βarrestin2-TEVp (nearby), and 3) 150 ng of P10.UAS-FLuc.~15 hr after transfection, cells re-plated in 96-well plates were exposed to furimazine or light conditions,

and ~9 hr later, UAS-FLuc reporter measurements were assayed as described above. In separate wells undergoing the same transfection and re-plating protocol, the total NanoLuc luminescence output was measured ~15 hr after transfection. Cells were exposed to 1.25 µL furimazine in 23.75 µL DMEM +10% FBS +20 mM HEPES in each 96-well, and luminescence was immediately analyzed at 25°C using a 1000 ms acquisition time, the Blue filter, and linear shaking for 3 s.

## SPARK2 drug screen

For SPARK2 drug screens, HEK293T cells plated directly on 6-well plates were transfected with 1) 350 ng of P6.pAAV-CMV-β2AR-eLOV-TEVcs-Gal4 or 350 ng of P4.pAAV-CMV-DRD1-eLOV-TEVcs-Gal4, 2) 100 ng of P7.NanoLuc-βarrestin2-TEVp, and 3) 150 ng of P10.pAAV-UAS-FLuc. DNA transfection and re-plating into 96-well plates were performed as described above ('*FLuc NanoLuc-SPARK experiments*'). Transfected cells were re-plated in 50 µL DMEM + 10% FBS in the 96-well plates. After ~15 hr, cells were exposed to 5 µM of a 92-drug compound sub-library (Biomol FDA-approved drug library, Stanford University High-throughput Bioscience Center, *Figure 4—source data 2*) and 10 µM furimazine for 1 hr (final volume in each 96-well was 100 µL). Two wells were exposed to a DMSO vehicle control, and two wells were exposed to either 5 µM isoetharine or 5 µM dopamine as positive spike-in controls. After 1 hr, the media was replaced with 100 µL of DMEM + 10% FBS, and the plates were wrapped in aluminum foil and incubated in the 37°C incubator for 9 hr prior to luminescence measurements on a plate reader. FLuc measurements were performed as described above ('*FLuc NanoLuc-SPARK experiments*').

## Direct NanoLuc-oβ2AR activation

The following lentiviral constructs were cloned: P12.pLX208-CMV-NanoLuc-oβ2AR-eLOV-TEVcs-Gal4, P13.pLX208-CMV-NanoLuc-βarrestin2-TEVp, and P14.pFPGW-UAS-Citrine. For lentiviral production of each construct, a T25 flask with ~90% confluent HEK293T cells was transfected with 2.5 µg of DNA plasmid, 0.25 µg of pVSVG, and 2.25 µg of delta8.9 lentiviral helper plasmid. DNA was mixed in 200 µL serum-free DMEM, then combined with 30 µL of PEI max and incubated at room temperature for 20 min. 4 mL of DMEM + 10% FBS was added to the mixture, and the entire volume was used to replace the media in the T25 flask. Cells were incubated for 48 hr at 37°C, and then the

**Table 1.** Plasmids used in this study.

| Name | Description | Vector-Promoter | Addgene |
|------|-------------|-----------------|---------|
| P1 | HA-β2AR-NNES-NanoLuc-15aa linker-AsLOV2(V416L) | pAAV-CMV | 125224 |
| P2 | gLuc sp-NanoLuc-HA-β2AR-NNES-AsLOV2(V416L) | pAAV-CMV | 125225 |
| P3 | mCherry-GSGS linker-Zdk1 | pAAV-CMV | 125226 |
| P4 | DRD1-NNES-eLOV-TEVcs-Flag-Gal4-V5 | pAAV-CMV | 125227 |
| P5 | AVPR2-NNES-eLOV-TEVcs-Flag-Gal4-V5 | pAAV-CMV | 104844 |
| P6 | β2AR-NNES-eLOV-TEVcs-Flag-Gal4-V5 | pAAV-CMV | 104841 |
| P7 | NanoLuc-15aa linker-βarrestin2-HA-GS linker-TEVp | pAAV-CMV | 125228 |
| P8 | UAS-Citrine | pAAV-UAS | 104839 |
| P9 | βarrestin2-HA-GS linker-TEVp | pAAV-CMV | 104845 |
| P10 | UAS-FLuc | pAAV-UAS | 104840 |
| P11 | IgK sp-HA-CD4-10aa linker-CIBN-NNES-NanoLuc | pAAV-CMV | 125229 |
| P12 | gLuc sp-NanoLuc-HA-oβ2AR-TS-eLOV-TEVcs-Flag-Gal4 | pLX208-CMV | 125230 |
| P13 | NanoLuc-15aa linker-βarrestin2-HA-GS linker-TEVp | pLX208-CMV | 125231 |
| P14 | UAS-Citrine | pFPGW-UAS | 125232 |
| P15 | gLuc sp-NanoLuc-15aa linker-ICAM-Nrxn3b-HA | pLX208-CMV | 125233 |
| P16 | gLuc sp-GS linker-HiBit-Flag-oβ2AR-TS-NNES-eLOV-TEVcs-Flag-Gal4-V5 | pAAV-CMV | 125234 |
| P17 | NanoLuc-15aa linker-βarrestin2-no HA-GS linker-TEVp | pAAV-CMV | 125235 |
| P18 | β2AR-NNES-NanoLuc-eLOV-TEVcs-Flag-Gal4-V5 | pAAV-CMV | 125236 |

DOI: https://doi.org/10.7554/eLife.43826.020

supernatant containing the secreted lentivirus was collected and filtered through a 0.45 μm syringe (VWR). Lentivirus was aliquoted and flash frozen in liquid nitrogen and stored at −80°C.

For experiments, HEK293T cells were grown to ~50% confluency in 48-well plates. The media for each well was replaced with 200 μL MEM +10% FBS, and the following volume of viruses was added to each well: 1) 100 μL Lenti-NanoLuc-oβ2AR-eLOV-TEVcs-Gal4, 2) 50 μL Lenti-NanoLuc-βarrestin2-TEVp, and 3) 60 μL Lenti-UAS-Citrine. Cells were returned to the 37°C incubator for 48 hr.

The media was then replaced in each well with 100 μL DMEM +10% FBS +20 mM HEPES with or without 50 μM 9-cis-retinal (Sigma). For the furimazine condition, 2.5 μL furimazine in 22.5 μL of media was added to each well (1:50) and cells were incubated at room temperature in the dark wrapped in aluminum foil for 20 min. For the light condition, cells were exposed to a white lamp (T5 Circline Fluorescence lamp, 25W, 6500K, 480 nm/530 nm/590 nm) for 10 min at room temperature. Following furimazine or light exposure, cells were washed with MEM +10% FBS and returned to the 37°C incubator for 24 hr wrapped in aluminum foil.

Cells were fixed in 4% paraformaldehyde at room temperature for 15 min, then washed 2x with DPBS. Citrine images were acquired at 10x magnification on the confocal and quantified using the CellSegm toolbox (*Hodneland et al., 2013*) in MATLAB.

### Trans-cellular NanoLuc-oβ2AR BRET

HEK293T cells plated directly on 6-well plates were grown to ~80% confluency. *Sender* cells were transfected with 1) 200 ng of P15.pLX208-CMV-NanoLuc-ICAM-Neurexin, while *receiver* cells were transfected with 1) 140 ng of P16.pAAV-CMV-oβ2AR-eLOV-TEVcs-Gal4, 2) 30 ng of P17.pAAV-CMV-NanoLuc-βarrestin2-no HA-TEVp, and 3) 15 ng of P8.pAAV-UAS-Citrine. For each well, plasmid DNA was mixed with 20 μL MEM and mixed with a solution of 20 μL MEM and 2 μL Lipofectamine reagent. The mixed solution was incubated for 10 min at room temperature, and then added directly to the cells and placed in the 37°C incubator. After 4 hr, cells were trypsinized, washed twice with DPBS, and re-plated onto fibronectin-coated 24-well plates at a 1:10 ratio of *receiver* cells to *sender* cells in 300 μL DMEM +10% FBS. Plates were wrapped in aluminum foil and incubated for 14 hr in the 37°C incubator.

For stimulation conditions, the media was replaced with 300 μL DMEM +10% FBS +20 mM HEPES with or without 50 μM 9-*cis*-retinal. For the furimazine condition, 3 μL furimazine was added to each well (1:100) and cells were incubated at room temperature for 20 min. For the light conditions, cells were exposed to a white lamp (T5 Circline Fluorescence lamp, 25W, 6500K, 480 nm/530 nm/590 nm) for 10 min at room temperature. After furimazone or light exposure, cells were washed with DMEM +10% FBS and returned to the 37°C incubator.

After 8 hr, cells were fixed with 4% formaldehyde in PBS for 10 min, then permeabilized with methanol at −20°C for 10 min. Cells were washed twice with DPBS and incubated in blocking buffer (5% Normal Donkey Serum, 0.02% sodium azide in 1x PBS) for 30 min at room temperature. Samples were then incubated with primary antibody (1:300 Mouse anti-Flag, Sigma F3165-1MG) in blocking buffer for 16 hr at 4°C. Samples were washed three times with DPBS and incubated with secondary antibody (1:1000 Goat anti-Mouse AlexaFluor 568, Life Technologies A11004) in blocking buffer for 4 hr at room temperature. Samples were washed three times with DPBS prior to imaging.

Images were collected at 20x or 60x magnification on a confocal microscope and analyzed in Fiji. For each cell that was Flag-positive, an ROI encompassing the cell was manually drawn and the reported ratio was calculated as (Citrine Intensity-Citrine background)/(AlexaFluor 568 Intensity-AlexaFluor 568 background) for each ROI. The background offset measurement was calculated from a region of the image containing no cells. Data were plotted/analyzed in MATLAB R2017a (Mathworks).

## Acknowledgements

We thank U Hochgeschwender (Central Michigan University, Michigan) for the plasmid containing NanoLuc and K Deisseroth (Stanford University, California) for the oβ2AR construct. We thank T Kirkland (Promega) for helpful discussions about NanoLuc. We thank B K Kobilka (Stanford University, California) for input designing and interpreting the results of the SPARK2 drug screen experiments.

## Additional information

### Funding

| Funder | Grant reference number | Author |
|---|---|---|
| Stanford University | Walter V. and Idun Berry Postdoctoral Fellowship | Christina K Kim |
| Chan Zuckerberg Biohub | CZ Biohub Investigator Program | Alice Y Ting |

The funders had no role in study design, data collection and interpretation, or the decision to submit the work for publication.

### Author contributions

Christina K Kim, Conceptualization, Data curation, Formal analysis, Funding acquisition, Validation, Investigation, Visualization, Methodology, Writing—original draft, Writing—review and editing; Kelvin F Cho, Min Woo Kim, Data curation, Formal analysis, Validation, Investigation, Visualization, Methodology, Writing—original draft, Writing—review and editing; Alice Y Ting, Conceptualization, Supervision, Funding acquisition, Methodology, Writing—original draft, Project administration, Writing—review and editing

### Author ORCIDs

Christina K Kim (iD) http://orcid.org/0000-0002-1466-7098
Kelvin F Cho (iD) http://orcid.org/0000-0003-2917-9713
Alice Y Ting (iD) http://orcid.org/0000-0002-8277-5226

### Decision letter and Author response

Decision letter https://doi.org/10.7554/eLife.43826.024
Author response https://doi.org/10.7554/eLife.43826.025

## Additional files

### Supplementary files

• Transparent reporting form
DOI: https://doi.org/10.7554/eLife.43826.021

### Data availability

All plasmids generated in this study (Table 1) have been deposited on Addgene. Source data for Figure 2, 3, 4, 5, 2-figure supplement 1, 3-figure supplement 1, 4-figure supplement 1, and 5-figure supplement 1 have been provided as Excel files.

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
