## [Decision Letter]

[Editors’ note: this article was originally rejected after discussions between the reviewers, but the authors were invited to resubmit after an appeal against the decision.]

Thank you for submitting your work entitled "Luciferase-LOV BRET enables highly specific transcriptional readout of cellular protein-protein interactions" for consideration by *eLife*. Your article has been reviewed by three peer reviewers, including Volker Dötsch as the Reviewing Editor and Reviewer #1, and the evaluation has been overseen by a Senior Editor.

Our decision has been reached after consultation between the reviewers. Based on these discussions and the individual reviews below, we regret to inform you that your work will not be considered further for publication in *eLife*.

All reviewers feel that your methods development projects provide very useful tools for the investigation of protein-protein interactions and that your current methods development offers new and improved possibilities. However, after discussion of the reviews we feel that the improvement – while useful – is too incremental for a publication in *eLife*. In particular a direct comparison with the original method to show the improvement is missing. In addition, an exciting application of the new method that provides new insight into an important biological problem would also have been beneficial. The detailed reviews are attached below.

*Reviewer #1:*

Kim et al. describe a further development of their recently published SPARK method for the identification of protein-protein interactions. In the previous method they had used external light to trigger a conformational change that allows a protease to cleave off a linker to a transcriptional regulator. One problem so far was a relatively high noise level. The authors now show that by combining the protease with a luciferase cleavage only occurs when the protein protein interaction is active, reducing the background signal. They show several applications for their method including cell-cell contacts.

Overall this is an interesting and very useful further development of their older method that is well documented.

*Reviewer #2:*

The authors describe refinement of a previously described method (SPARK) to couple simultaneous light application and protein-protein interaction (PPI) to drive transcription. In this iteration, a luciferase domain is added, enabling both substitution of light application with luciferin application (if desired) and reduced background as a result of the steep distance dependence of BRET from the luciferase. They demonstrate that this system works in HEK cell cultures for multiple PPIs. They also show an application using BRET from the luciferase to β2AR to drive transcription as a result of cell-cell contact.

The instantiation of a nested molecular logic gate to decrease background is a creative work of protein engineering. The work appears to be carefully done and is presented clearly. My enthusiasm for this manuscript and tool is limited by the lack of functional demonstration beyond the reduced preparation of HEK cells in a dish. The complexity of the system (many protein domains/constructs, multiple gating mechanisms) makes it seem unlikely to me that this would be desirable for an in vitro high-throughput screening application. The temporal gating with light or luciferin suggests that the intended use would be intact in vivo preparations, but they don't show any data suggesting that in vivo use is possible.

*Reviewer #3:*

This manuscript by Kim and coworkers details the generation and testing of "SPARK2", a new tool for evaluating cellular protein/protein interactions (PPIs). Taking advantage of the light-dependent conformational changes of LOV domains (and a prior "SPARK" tool by the same group, here renamed "SPARK1"), the authors here develop a transcriptional reporter system which couples the PPI-dependent recruitment of a luciferase-protease fusion hybrid protein to its partner to the subsequent cleavage and release of a transcriptional activator. Of note, this iteration is substantially improved over the prior SPARK by using the luciferase-enabled BRET activation of LOV photochemistry vs. prior illumination-dependent triggering, both improving signal-to-noise and enabling use in HTS-type instrumentation without easy illumination capabilities. The authors exhibit these favorable characteristics in a variety of different formats, most impressively by facilitating the detection of intercellular contacts between a luciferase-tagged protein on one cell to an adjacent cell containing an engineered opto-triggered GPCR.

In aggregate, I find this manuscript to present some interesting improvements on the SPARK1 tool but with enough concerns to substantially lower my enthusiasm for publication here. Most fundamentally, the improvements here seem to be somewhat expected by a). Proshkina et al., 2018's report that NanoLuc is sufficient to activate LOV photochemistry within the LOV-derived miniSOG proteins and b). the common improvements in S/N expected from "AND" type gates widely in synthetic biology. Several number of points made throughout the manuscript are similarly quite expected (e.g. compatibility of NanoLuc and FLuc) or demonstrated multiple times (e.g. demonstrations that NanoLuc needs to be proximal to LOV domain). Finally, the surprising lack of a direct comparison between SPARK1 and SPARK2 and substantial number of minor technical points (described below) also contributed to concerns that readers would not have some important information presented to them needed to determine the relative utilities of these tools vs. existing alternatives. In aggregate, I suggest that these issues outweigh the strengths of the work to the point that publication elsewhere is recommended.

---

## [Author Response]

[Editors’ note: the author responses to the first round of peer review follow.]

Reviewer #2:[…] The instantiation of a nested molecular logic gate to decrease background is a creative work of protein engineering. The work appears to be carefully done and is presented clearly. My enthusiasm for this manuscript and tool is limited by the lack of functional demonstration beyond the reduced preparation of HEK cells in a dish.

Thank you for your appreciation of the nested logic gate design. Regarding the reviewer’s concern, we point out that there is a broad literature, and indeed entire fields of biology, dedicated to the detection of protein-protein interactions (PPIs) in human cell in vitro preparations. The are many advantages of performing these experiments in a dish rather than in vivo – less variability in expression levels of the tool, precise control over the timing and delivery of drugs inducing the PPI, and of course higher-throughput (demonstrated below).

The complexity of the system (many protein domains/constructs, multiple gating mechanisms) makes it seem unlikely to me that this would be desirable for an in vitro high-throughput screening application.

We agree with the reviewer that the number of protein domains/constructs presents a general challenge for implementing SPARK, or other similar PPI-detection tools with the same number of protein fusion constructs (e.g., TANGO (Barnea et al., 2008), synNotch (Morsut et al., 2016)). Because it is difficult to control the relative expression ratios of these multiple constructs, we often observe high amounts of background reporter expression even in the absence of a true PPI using SPARK1. We find that the multiple gating mechanisms that we implemented with SPARK2 circumvent this problem rather than exacerbate it. Across multiple biological replications and different experimenters, we routinely find that SPARK2 results in no additional detectable background in the absence of a PPI. Because the assay is now so robust, we designed and executed a high-throughput drug screen for GPCR activation in order to demonstrate that our platform is desirable and easy to implement for in vitro screening applications. We observed robust detection of compounds that are known to activate β2AR and DRD1, in addition to unexpected compounds that were found to activate DRD1 (e.g. a known DRD2 antagonist, Haloperidol; Figure 4C-E). Furthermore, while we expected dopamine to activate β2AR as a partial agonist, instead we observed no SPARK2 activation, suggesting that dopamine could partially activate β2AR via an arrestin-independent mechanism. These results directly demonstrate the potential for new biological insight using SPARK2.

The temporal gating with light or luciferin suggests that the intended use would be intact in vivo preparations, but they don't show any data suggesting that in vivo use is possible.

The temporal gating with light or luciferin, rather than intended to enable intact in vivopreparations, is instead essential for lowering background expression to enable highly specific and sensitive detection of PPIs in vitro. While we demonstrated the necessity of temporal gating extensively in our original SPARK manuscript, given the structure of the Research Advance article, we did not go into detail about the utility of light-gating for in vitro PPI-detection. We have re-emphasized this point in the first paragraph of the Introduction, and explicitly direct the readers back to the original SPARK manuscript as well. We thank the reviewer for highlighting these important points that we have now addressed via new experiments and textual clarifications.

Reviewer #3:[…] In aggregate, I find this manuscript to present some interesting improvements on the SPARK1 tool but with enough concerns to substantially lower my enthusiasm for publication here. Most fundamentally, the improvements here seem to be somewhat expected by (a) Proshkina et al., 2018's report that NanoLuc is sufficient to activate LOV photochemistry within the LOV-derived miniSOG proteins and (b) the common improvements in S/N expected from "AND" type gates widely in synthetic biology.

Thank you for your concerns. However, we respectfully object to the idea that the improvements enabled here were obvious, or even somewhat expected. We address both points (a) and (b) with our own experimental attempts to replicate the Proshkina et al., 2018 paper, and with a new experiment to illustrate the distinction between our nested “AND” gate design, and the traditional single “AND” gates commonly used in synthetic biology.

a) Proshkina et al., 2018, reports that NanoLuc-miniSOG BRET can induce phototoxicity and cell death via the production of reactive oxygen species. We agree with the reviewer that this study is relevant to ours, and have cited it prominently in the last paragraph of the subsection “Selection of NanoLuc as the luciferase donor for LOV”, and in the second paragraph of the Discussion. However, we do not agree that Proshkina et al., 2018, makes our results predictable or obvious. We are employing intermolecular, PPI-driven BRET between NanoLuc and LOV, rather than intramolecular BRET via direct NanoLuc-miniSOG fusion (which should be much more efficient). We are using a much more precise and molecularly specific readout of BRET (reporter gene transcription) than cell death. miniSOG and LOV domains have different action spectra and photophysical properties. For all these reasons, the Proshkina result does not guarantee that our SPARK2 design would be successful.

The second point concerns the robustness of the Proshkina et al. result, particularly for live-cell assays in HEK293T cells. We were concerned when we read that their experiment required treating NanoLuc-miniSOG-expressing cells with 43-75µM furimazine over 48 hours (by comparison, SPARK2 uses 10µM furimazine for 20 minutes). Hall et al., 2012 previously showed that furimazine concentrations must be kept to 10µM or less over 2 hours to be non-lethal to cells. The same first author of the Proshkina et al., 2018 paper published a different paper around the same time that documented furimazine-dependent toxicity in the same cell-line (SK-BR-3) used in their NanoLuc-miniSOG BRET photoxicity paper (Shipunova et al., 2018). They reported ~50% cell death in untransfected SK-BR-3 cells exposed to 20µg/mL of furimazine (52µM) for 48 hours. We attempted to reproduce their experiment ourselves in HEK293T cells, and found no difference in cell viability when comparing untransfected cells to NanoLuc-miniSOG-expressing cells, upon treatment with 75µM furimazine. Our inability to reproduce the key observation of the Proshkina paper (albeit in HEK rather than SK-BR-3 cells) further underscores that their study does not diminish the novelty of our work.

b) We agree with the reviewer that single “AND” gates commonly improve S/N in the field of synthetic biology. Indeed, this was the main point of our original SPARK1 publication (Kim et al., 2017), which implemented an “AND” gate using light + PPI to achieve improved S/N compared to its predecessor, TANGO (Barnea et al., 2008). However, the starting point for the present study was the observation that this simple “AND” gate design was not adequate for detecting cellular PPIs with high specificity across a range of conditions and expression levels (i.e., tool is not robust). This is why we moved on to design a different logic gate, one based on *two* nested “AND” gates.

We could have achieved an all-drug-gated version of SPARK by simply fusing NanoLuc directly to the LOV domain, rather than to the arrestin-TEVp. This would mimic a single “AND” gate, in which LOV uncaging by luciferase, plus a PPI-induced interaction between TEVp and TEVcs mediate the release of the transcription factor. In this case, however, we would not expect any improvement in S/N, as the TEVp could still cleave the TEVcs in the absence of a PPI whenever furimazine is present (as in Figure 1C). However, in the *nested* “AND” gate design, even if furimazine is present, the LOV domain will remain caged unless the PPI is also present to bring the NanoLuc-arrestin-TEVp within proximity of the LOV. This nested design is elegantly simple yet effective, requiring *no additional constructs* compared to the single “AND” gate configuration.

To help make the rationale for this design more clear, we present new data that compares SPARK2 to an alternative design in which NanoLuc is fused directly to LOV. Figure 3E-H show that this alternative design (with intramolecular, PPI-independent NanoLuc-LOV BRET) still results in background signal (in the absence of a true PPI), while SPARK2 with the nested “AND” gate has no additional detectable TEVcs cleavage in the absence of the PPI – even when the furimazine is left in the media for the entire ~8 hour incubation period. Thus, it is not simply the phenomenon of NanoLuc-LOV BRET that is necessary, but the PPI-induced proximity between NanoLuc and LOV that is crucial for eliminating background TEVcs cleavage with SPARK2. The striking difference and improvement we observed with the nested “AND” gate design make us hopeful that it will enable new synthetic biology logic designs, especially when a single “AND” gate is found to have insufficient gating due to variable protein expression in living cells.

For these reasons above, we ask reviewer 3 to reconsider their judgement that the application of PPI-induced NanoLuc-LOV BRET lacks novelty.

Several number of points made throughout the manuscript are similarly quite expected (e.g. compatibility of NanoLuc and FLuc) or demonstrated multiple times (e.g. demonstrations that NanoLuc needs to be proximal to LOV domain).

We felt it was important to thoroughly demonstrate that our tool works robustly across several different contexts – e.g. that proximity-dependence is maintained, and that compatibility with existing luciferase reporters is maintained. We hope the reviewer can agree that while the results could be expected by an expert who has a deep understanding of the LOV domain photochemistry, they remained to be proven to the broader scientific community in a detailed manner. For example, the very reason we demonstrated proximity-dependence multiple times is that this was a consistent concern raised when we presented our work to broader scientific audiences, many of whom were not protein-engineers but instead biologists and potential users of the tool.However, we note that we have moved the proximity-dependence demonstration for SPARK2 into a supplemental figure (Figure 4—figure supplement 1A-B), as we now include new data in the main Figure 4 demonstrating a biological application of a high-throughput drug screen for β2AR and DRD1 activation (Figure 4C-E).

Finally, the surprising lack of a direct comparison between SPARK1 and SPARK2 and substantial number of minor technical points (described below) also contributed to concerns that readers would not have some important information presented to them needed to determine the relative utilities of these tools vs. existing alternatives. In aggregate, I suggest that these issues outweigh the strengths of the work to the point that publication elsewhere is recommended.

We have included a side-by-side comparison of SPARK1 and SPARK2 in the revised manuscript (Figure 3—figure supplement 1A-B).The reason we did not do so before was that SPARK2 with light-gating is essentially the same as SPARK1 with light-gating (thus the original submission compared SPARK2 + furimazine to SPARK2 + light, rather than SPARK2 + furimazine to SPARK1 + light).

We ask that the reviewer please reconsider judgements of novelty and expectation given the stated goals and requisites of an *eLife* Research Advance. We appreciate the reviewer’s thoughtful suggestions, which we feel have strengthened the manuscript and highlighted the advantages of SPARK2 over SPARK1.